# High-Pressure-Induced Sublethal Injuries of Food Pathogens—Microscopic Assessment

**DOI:** 10.3390/foods10122940

**Published:** 2021-11-30

**Authors:** Justyna Nasiłowska, Aleksandra Kocot, Paulina Natalia Osuchowska, Barbara Sokołowska

**Affiliations:** 1Wacław Dąbrowski Institute of Agricultural and Food Biotechnology—State Research Institute, 01-142 Warsaw, Poland; barbara.sokolowska@ibprs.pl; 2Department of Immunology and Food Microbiology, Institute of Animal Reproduction and Food Research of the Polish Academy of Sciences, 10-748 Olsztyn, Poland; a.kocot@pan.olsztyn.pl; 3Biomedical Engineering Centre, Institute of Optoelectronics, Military University of Technology, 01-142 Warsaw, Poland; paulina.osuchowska@wat.edu.pl; 4Institute of High Pressure Physics, Polish Academy of Sciences, Laboratory of Biological Materials, 29/37 Sokołowska str., 01-142 Warsaw, Poland

**Keywords:** HHP, foodborne pathogens, SEM, TEM, EFM

## Abstract

High Hydrostatic Pressure (HHP) technology is considered an alternative method of food preservation. Nevertheless, the current dogma is that HHP might be insufficient to preserve food lastingly against some pathogens. Incompletely damaged cells can resuscitate under favorable conditions, and they may proliferate in food during storage. This study was undertaken to characterize the extent of sublethal injuries induced by HHP (300–500 MPa) on *Escherichia coli* and *Listeria inncua* strains. The morphological changes were evaluated using microscopy methods such as Scanning Electron Microscopy (SEM), Transmission Electron Microscopy (TEM), and Epifluorescence Microscopy (EFM). The overall assessment of the physiological state of tested bacteria through TEM and SEM showed that the action of pressure on the structure of the bacterial membrane was almost minor or unnoticeable, beyond the *L. innocua* wild-type strain. However, alterations were observed in subcellular structures such as the cytoplasm and nucleoid for both *L. innocua* and *E. coli* strains. More significant changes after the HHP of internal structures were reported in the case of wild-type strains isolated from raw juice. Extreme condensation of the cytoplasm was observed, while the outline of cells was intact. The percentage ratio between alive and injured cells in the population was assessed by fluorescent microscopy. The results of HHP-treated samples showed a heterogeneous population, and red cell aggregates were observed. The percentage ratio of live and dead cells (L/D) in the *L. innocua* collection strain population was higher than in the case of the wild-type strain (69%/31% and 55%/45%, respectively). In turn, *E. coli* populations were characterized with a similar L/D ratio. Half of the cells in the populations were distinguished as visibly fluorescing red. The results obtained in this study confirmed sublethal HHP reaction on pathogens cells.

## 1. Introduction

In recent years, the development of microscopic methods has revolutionized the world of science. Microscopic techniques have started to be used for bacterial observation as a complement to scientific research and conventional diagnostic tests [1]. Nowadays, they are applied in the areas of clinical pathology, food, and water quality, where biological detection and quantification are significantly important [2,3]. Moreover, they are widely used to analyze the physiological state of microorganisms involved in biotechnological processes carried out on an industrial scale, where stress factors often appear [4]. Stress factors interfere with the growth and proper metabolism of bacteria, which causes a decrease in vitality and the inhibition of metabolic pathways [5]. The routine detection performed by counting CFU [colony forming unit] started to be replaced by instrumental techniques, such as microscopy, flow cytometry, optical methods, and bioluminescence [3]. In the world of biological science, Electron Microscopy (EM) has played the key role. The rapid improvement of EM has enabled advances in research into cell morphology and has provided detailed structural information and cell components such as membranes, genomes, and ribosomes. Until recently, it was believed that the bacterial cell is an amorphous structure composed of various macroparticles [6]. EM has demonstrated that the bacterial cell has a precisely defined organization. The enlargement of the limit of resolution in EM has allowed one to observe bacteria in detail. EM demonstrated that bacteria have a cytoskeletal structure with a circular nucleoid with no nuclear envelope [7]. Moreover, EM provides information on topography, morphology, including pathological alterations of the ultrastructure, and composition of the sample. The first images of bacteria were published after the transmission electron microscope (TEM) was introduced. SEM offers the possibility to inspect the cell surface and confirm morphological damage to the cell wall [8]. In turn, TEM gives opportunities to visualize the cellular ultrastructure, which is crucial in understanding of how cells and tissues function in both normal physiological and pathological states. In addition, in biological science, both of them are often used for identifying new bacteria and virulent strains, uncovering new species, the study of bacterial adhesion [8,9], performance the bacterial membrane injury test after high-pressure treatment [10,11], pulsed electric fields [12], and others treatments [13]. SEM is also compatible with immunolabeling techniques to label specific features on the surface of cells [8].

Epifluorescence microscopy (EFM) is another useful type of microscopy. It allows the assessment of the activity of numerous determinants of cellular activity, and not just the ability to multiply and grow on culture media [14,15]. EFM is commonly used in the study of the physiological state of bacterial cells. Various parameters of the physiological state of bacterial cells can be labeled, depending on the used indicators [16]. These indicators are fluorescent dyes, known as fluorochromes or fluorophores [17]. Various types of fluorescent dyes are ascertained and inform about cell activity determinants such as membrane integrity, pump activity, membrane potential, or metabolic activity [18]. There are also commercial kits that contain selected dyes. These dyes allow the quick labeling and differentiation of cells in terms of selected parameters. An example of such a dye system is the Backlight LIVE/DEAD vitality kit (Termo Fisher Scientific, Walham, MA, USA). The abovementioned kit contains two dyes—Syto^®^ 9 (Termo Fisher Scientific, Walham, MA, USA) and propidium iodide (PI) (Termo Fisher Scientific, Walham, MA, USA). Both dyes are markers of nucleic acids; however, they have a different molecular weight. They differentiate cells into living or dead cells, based on the intact or damaged cytoplasmic membranes. The Syto^®^ 9 has a low molecular weight and penetrates both living and dead cells, emitting green fluorescence. On the other hand, PI has a high molecular weight and penetrates cells only with damaged cytoplasmic membranes, giving red fluorescence [19]. Hence, in the microscopic image, green and red cells can be observed, corresponding to living and dead cells, respectively. Previous studies have shown the effectiveness of fluorescent staining combined with microscopic analysis (EFM). They can be noticed in assessing the physiological condition of bacterial cells, subjected to stress conditions, such as osmotic stress [20], heat stress [21], the use of disinfectants [22], or high pressures [23].

Microscopic analytical methods might be important especially for testing new alternative preservation treatments, such as High Hydrostatic Pressure (HHP). HHP has become a widely used way of food preservation in many countries in the world. Nowadays, it belongs to the group of leading innovative technologies, which guarantee food assets desired by consumers [24,25,26,27,28]. Food and nutrition scientists have shown that HHP preservation brings various benefits regarding food quality and safety. HHP maintains physical–chemical food properties that guarantee flavor, color, and product composition [29,30,31,32]. Moreover, it provides high microbiological safety, such as thermal sterilization [33,34,35]. Despite the above benefits, HHP technology is still the subject of research interest, especially for microbiologists. HHP can result not only in the significant inactivation of microorganisms, but also induces a wide range of sublethal injuries in the cell population [36,37]. The magnitude of these phenomena may be diverse depending on the genus or even species of microorganism. Additionally, the range of injuries depends on both the physiological condition of the bacterial strain and pressure process parameters, as well as on the intrinsic properties of the preserved food [36,38]. It was determined that the most likely first target of pressure is cytoplasmic membranes. As a result, there is a loss of plasma integrity, deformation of the membrane structure, and disruption of ion exchange in the bacterial cell. Moreover, pressure may influence the spatial organization of the cell. It damages the genetic mechanism and leads to unfavorable biochemical reactions, changes in ribosome conformation [39,40,41]. HHP-sublethal injury incurs cell functional disorders, which may be transient or permanent. On the one hand, the sublethally injured cell is characterized by the reduction in growth rate or the inability of growth in standard laboratory media. On the other hand, adaptations of microorganisms to sublethal stress initiate a range of responsive strategies within the bacterial cells [42]. Cells remain alive but undetectable. Moreover, they can repair themselves and proliferate in foods under appropriate conditions. As a consequence, it can result in a potential hazard that endangers food safety [28,43]. Therefore, the existence of sublethal damage of bacterial cells is the key issue that may compromise the efficiency of food preservation technologies. Thus far, extensive research on the effectiveness of HHP on bacteria has already been carried out. However, detailed descriptions of the responses of HHP-sublethally injured bacteria have not been available until now [28,37,38]. Scientists have suggested that both cell injury and the tailing effect are some of the factors that should be elucidated for improving the efficiency of new technologies. Moreover, the role of cell structure, physiology, and gene regulation in microbial resistance to alternative preservation technologies should also be investigated [43].

This study aimed to focus on the detection of sublethal injuries of *Escherichia coli* and *Listeria innocua* strains, triggered by HHP. For this purpose, various microscopy techniques were used. A detailed analysis of sublethal injuries cell-by-cell was carried out. The visualization of changes in cell morphology features was performed with electron microscopies—SEM and TEM. The second step of this work was to evaluate the physiological state, regarding the whole population using EFM and the LIVE/DEAD Bacterial Vitality Kit. This comprehensive approach enabled the evaluation of several parameters, which indicate sublethal injuries. The results correlation obtained with all methods was evaluated. Moreover, the abovementioned analysis was carried out to obtain a better understanding of the bacterial cells’ response to mild HHP.

## 2. Materials and Methods

### 2.1. Preparation of Bacterial Suspensions and Culture Conditions

*Listeria innocua* (CIP80.11T) obtained from the Culture Collection of the Institute Pasteur (Paris, France), *Escherichia coli* (ATCC 7839) obtained from American Type Culture Collection (Manassas, VA, USA), and two wild isolates (wild-type strains) from unpasteurized commercial Polish beetroot juice *Listeria innocua* 23/13 and *Escherichia coli* 61/14, obtained from our own culture collection of the Department of Fruit and Vegetable Product Technology at IAFB (Warsaw, Poland), were used in this study. Each strain was stored before use in a Cryobank (temperature below—27 °C ± 3 °C). A pure culture in the form of immobilized sterile beads was added to a tube containing 10 mL of Brain Heart Infusion (BHI) broth (BioMerieux, I’Etoile, France). Then, bacterial subcultures were incubated under static conditions at 37 °C for 24 h. To prepare the second subculture, each overnight culture was moved with a 10 µL loop on a Petri dish, using plate count analytical methods with appropriate agar, and incubated. For each kind of species, *E. coli* and *L. innocua* Tryptic Soy (TSA) agar (Biocar Diagnostics, Beauvais, France) or Tryptic Soy Yeast Extract (TSYE) agar (Biocar Diagnostics, Beauvais, France) were used, respectively. After that, grown colonies were moved from the plate by the 10 µL loop to 250 mL Erlenmeyer flasks containing 200 mL of the appropriate broth: Tryptic Soy Broth (TSB) (Biocar Diagnostics, Beauvais, France), or Tryptic Soy Broth with Yeast Extract (TSBYE) (Biocar Diagnostics, Beauvais, France), respectively. Subcultures were incubated under the same abovementioned conditions until the time of obtaining the stationary phase. Consecutive subcultures were prepared by the addition of 10 mL of the previously mentioned subculture into fresh 200 mL broth. Finally, 200 mL aliquots were taken from the cultures and centrifuged (4000× *g*, 10 min, 4 °C) (Rotina 380R Hettich Instruments, Tottlingen, Germany). Subsequently, supernatants were removed. The sedimented cells were resuspended in phosphate-buffered saline (PBS, pH 7.2) and centrifuged one more time. The washing procedure was repeated threefold. Model suspensions of tested bacteria were prepared in PBS in 1:9 (*v/v*). The initial concentration of inoculum was about 7 log CFU/mL.

### 2.2. HHP Device and Parameters

Tested samples were transferred into sterile polyethylene tubes (Sarstedt, Newton, MA, USA) in 13 mL portions in duplicate. Then, samples were exposed to high hydrostatic pressures using a U 4000/65 device (Unipress, Warsaw, Poland). The maximum volume of the treatment chamber was 0.95 L, and the maximum working pressure was 600 MPa. The HHP device worked in the range of temperatures from −10 °C to +80 °C. The pressure-transmitting fluid that was used was distilled water and polypropylene glycol (1:1, *v/v*). The time needed to obtain the pressure up to 400 MPa was about 75 s. The release time was 2–4 s. Due to the adiabatic heating, the temperature increased approximately 3 °C per 400 MPa. The pressurization times reported did not include the come-up and come-down times. Samples were subjected to HHP at two variants of parameters: 400 MPa/5 min for *L. innocua* strains and 500 MPa/5 min for *E. coli* strains, at an ambient temperature (i.e., approximately 20 °C). The choice of process parameters for this study was deliberate. The main aim of this study was the detection of sublethal injuries. Thus, we chose the optimal pressure and time duration, based on our previous studies, to induce sublethal injuries of tested strains and not to inactivate them. For instance, our previous data showed that a pressure application of 400 MPa up to 10 min did not trigger significant injuries on both tested *E. coli* strains, while *L. innocua* strains were not detected. Both results were obtained with plate count analytical methods. The experiment in this study was performed twice with two independent repetitions for each trial (*n* = 4). Unpressurized samples were used as a control. After exposure to pressure, samples were analyzed with the usage of microscopic methods.

### 2.3. Scanning Electron Microscope Protocol (SEM)

To visualize bacterial morphology, the 10^7^ CFU/mL density suspension of bacteria was filtered through 0.22 µm PVDF membrane filters (Isopore^TM^, Millipore, Ireland). After that, bacteria on the membrane were fixed with 4% paraformaldehyde (MerckMillipore, Burlington, MA, USA) and 2% glutaraldehyde (MerckMillipore, Burlington, MA, USA) in PBS at 4 °C overnight. Thereafter, bacteria were washed three times with distilled water and incubated in 1% osmium tetroxide (Sigma-Aldrich, St. Louis, MO, USA) at 4 °C for 16 h. The specimens were dehydrated in consecutively increasing concentrations of ethanol (30, 50, 70, 90, 96, and 99%) and acetone (30%, 50%, and 100%). Bacteria on the filter were dried in a critical point dryer (Leica EM CPD 300, Wetzlar, Germany) and were then coated with platinum using a sputter coater (Leica EM ACE200, Wetzmar, Germany). The morphology of bacteria was analyzed using a scanning electron microscope STEM (Quanta FEG450, FEI, OR, USA) at 10 kV (spot 3.5) in a high vacuum mode with an Everhart–Thornley Detector (ETD).

### 2.4. Transmission Electron Microscopy Protocol (TEM)

Tested samples were fixed with 2.5% glutaraldehyde cacodylic buffer and incubated for 1 h. After that, samples were washed with 0.1 M of cacodylic buffer. The next step was to postfix the samples in 1% OsO_4_ in ddH_2_O for 1 h and wash them three times in ddH_2_O. After postfixation, the samples were dehydrated through a graded series of EtOH (30%—10 min, 50%—10 min, 70%—24 h, 80%—10 min, 90%—10 min, 96%—10 min, anhydrous EtOH—10 min, acetone—10 min). After that, they were infiltrated with Epon resin in acetone (1:3—30 min, 1:1—30 min, 3:1—2 h). Moreover, they were infused twice for 24 h in pure Epon resin and polymerized at 60 °C for 24 h. Next, 60 nm sections were prepared using a RMC ultramicrotome MT-X (RMC Boeckeler Instruments, Tucson, AZ, USA) and contrasted with uranyl acetate and lead citrate according to Reynolds (1963) [44] Samples were examined on a LIBRA 120 electron microscope produced by Zeiss (Oberkochen, Germany). Finally, images were captured with the Slow Scan CCD camera (Proscan) using EsiVision Pro 3.2 software (Soft Imaging Systems GmbH, Münster, Germany). Measurements were performed using the analySIS^®^ 3.0 image-analytical software (Soft Imaging Systems GmbH, Münster, Germany).

### 2.5. Epifluorescent Microscopy (EFM)

An amount of 1 mL of each sample was centrifuged, and the pellet was resuspended in 1 mL of sterile phosphate-buffered saline (PBS, pH 7.2). Each sample was stained with a Live/Dead BacLight^TM^ viability kit (Termo Fisher Scientific, Walham, MA, USA) according to the manufacturer’s recommendations [45], and incubated at ambient temperature for 15 min in darkness. After incubation, samples were filtered so that the cells settled on the surface of the black polycarbonate filters (Ø 13 mm, 0.2 µm; IsoporeTM, Millipore, Darmstadt, Germany). The filtration set consisted of polypropylene Millipore Swinnex^®^ membrane filter holders (MerckMillipore, Burlington, MA, USA) and a sterile medical needle and syringe. The filters were placed on microscopic slides, allowed to air-dry, and covered by a coverslip with mounting oil (Termo Fisher Scientific, Walham, MA, USA). The microscopic analysis was performed with an epifluorescence microscope Nikon E800 (Nikon Instruments Europe BV, Amsterdam, The Netherlands). The set of filters for green and red fluorescence were used. For green fluorescence, the following were used: excitation filter 465–495 nm and emission filter 515–555 nm. For red fluorescence, the following were used: excitation filter 540–580 nm and emission filter 600–660 nm. All of them correspond to the colors emitted by the Syto^®^ 9 and PI. Visualization was made with the photo collecting software Lucia G version 4.82 (Laboratory Imaging, Prague, Czech Republic). Then, representative images were merged with the usage of Image J software. The calculation of the surface area of live and dead cells was made with the usage of QuPath software 3.0.3 version (GitHub, San Francisco, CA, USA).

### 2.6. Statistical Analysis

Statistical analysis of the results was performed by the two-way ANOVA statistical model with Tukey’s test, using Statistica version 13 (TIBCO Software Inc., Palo Alto, CA, USA). The differences were considered significant at *p* < 0.05. A statistical comparison was made for the results, obtained for live and dead cells for *L. innocua* and *E. coli* strains separately.

## 3. Results and Discussion

### 3.1. Escherichia coli EMs Observations

The spatial organization of *E. coli* strains, collection, and wild-type strain, before and after HHP, is demonstrated by the TEM and SEM images in Figure 1 and Figure 2, respectively. The micrographs of both untreated samples showed characteristic rod-shaped *E. coli* cells, single or dividing. Morphologically, *E. coli* has a double cell wall, with a thin inner wall of peptidoglycan and an outer wall of carbohydrates, proteins, and lipids. TEM and SEM images confirmed a clearly defined, intact cell membrane and cell wall with a rough surface. Cells contained centrally located genomes surrounded by the integrated cytoplasmic area and tightly packed ribosomes. The application of pressure resulted in morphological differences in the appearance of both tested strains. In some bacterial cells, a loss of the general cellular shape was spotted by SEM. Individual cells were collapsed or even gutted and were characterized with squeezed envelopes. However, there was no significant membrane damage observed. Internal structural cell changes were visualized by TEM. The compression of interior regions and the expansion of nucleoid regions were observed in the whole population. This indicates an aggregation of cytoplasm in the amorphous region and disorganization of the genome area containing fibrillar regions. The abovementioned intracellular disruption was significantly more extensive in cells of the wild-type strains. Moreover, blank spaces in the cytoplasm were observed (Figure 2). A similar effect of the internal cell disruption of *E. coli* strains, visualized by TEM, was reported by Prieto-Calvo et al. [11]. Two strains, pathogenic *E. coli* VTEC O157:H7 and nonpathogenic *E. coli*, were treated by HHP for 5 min under 300 MPa and 600 MPa. The changes in the molecular composition occurred within the cytoplasm and genome area. However, contrary to these findings, we observed neither cellular enlargement nor winding shapes. The same ultrastructural modification profile of *E. coli* K-12TG1 induced by HHP (150 MPa, 250 MPA, and 350 MPa at 25 °C) was detected. The condensed nucleoids and aggregated proteins occurred in the whole population; however, the intensity of structural changes increased along with the extension of pressure [46]. Other scientists demonstrated that a pressure below 300 MPa had no impact on the spatial organization of *E. coli* ATCC 25922 cells. The cell, assessed by TEM, maintained a distinct membrane and cell wall. The most visible changes were noticed when the pressure increased to 500 MPa. The aggregation of cytoplasmic material, enlargement of electron-transparent ranges in the cell cytoplasm, and disruption of cell membrane appearance, including the breakdown of the peptidoglycan layer, occurred. After 30 min of pressure treatment, expanded nucleoid regions and compacted interior regions were observed, which correspond to our findings obtained after 5 min of treatment [47]. A similar phenomenon was observed under TEM for *Salmonella enterica* serovar Thompson. *Salmonella* was characterized with amorphous compacted regions, probably representing denatured cytoplasmic proteins after 250 MPa for 10 min. The increasing pressure up to 500 MPa resulted in extreme condensation of the cytoplasm, whereas the outline of the cells was intact [10]. Hsu et al. [48] investigated six strains of *E. coli* “Big Six” non-O157 STECs and five strains of *E. coli* O157:H7 in fresh strawberry puree under HHP. The results showed the rapture of cell structures for tested bacteria. SEM micrographs showed that pressurization at 350 MPa for 15 min contributed to the envelopes’ damage and collapsed off the edge of *E. coli* cells. The extension of pressure up to 550 MPa resulted in the severe injury of bacterial cells. It is known that Gram-negative bacteria are more sensitive to pressure than Gram-positive cells are. However, variations in pressure resistance occur even among strains belonging to the same species [49]. In this study, we analyzed the response of Gram-negative bacteria to pressure treatment below 600 MPa in ambient temperature. In most instances of quoted data, the type of morphological changes of Gram-negative bacterial cells induced by pressure was similar. However, the extension of injuries depends on the strain, suspension medium, and device’s technical parameters.

### 3.2. Listeria innocua EMs Observations

TEM and SEM snapshots of both strains of *L. innocua*, before and after HHP, are presented in Figure 3 and Figure 4, respectively. Morphologically, *Listeria* sp. has a thick cell wall made of peptidoglycan. Images of control samples showed the regular, rod-shaped cells with the continued, smooth, distinct cell membrane and uniform cell cytoplasm. The interior of the cells was properly organized with a centrally located genome and tightly packed ribosomes. Some cells were divided into both populations of *L. innocua*. Pressure application triggered minor and major changes in internal cell organization in both strains, visualized by TEM. The most notable changes were observed for the genome area. The significant disorganization of nucleic acid was confirmed. Contrary to *E. coli* strains, cytoplasm aggregation was reported not in the whole population, but exclusively for several cells of the tested *L. innocua* wild-type strain. Moreover, membrane permeabilization was also observed. In turn, the collection strain was characterized with an intact cell membrane. However, both *L. innocua* visualized by SEM were distorted and dilacerated. The deprivation of the membrane integrity revealed a discontinuous and distorted appearance and released its intracellular contents. Presumably, the disjunction between SEM and TEM visualization of the *L. innocua* collection strain could have been caused, not due to pressure, but probably because of sample preparation. Scientists claimed that the drying step is a common problem, especially for wet biological specimens [2]. Complementary observations by TEM in the spatial organization of *Listeria monocytogenes* NCTC 7973 suspended in TSB were reported by Mackey et al. [10]. Pressure application under 250 MPa at ambient temperature triggered significant changes. The interior of the cell was disrupted. Clear unusual symmetrical areas within the cytoplasm, which resembled gas bubbles, were spotted. Moreover, the cytoplasm was devoided by ribosomes. It was suggested that membrane invaginations might have been caused due to the osmotic effects or phase changes in the membrane [10]. In turn, exposure to a pressure of 500 MPa caused an increase in the number of cells with clear vacuolar regions, additionally containing fibrillar regions of DNA. Another study conducted by Huang et al. [49] with *Listeria monocytogenes* BCRC 15354 in milk was carried out with the usage of both electron microscopies, SEM and TEM. The results showed that after treatment with 450 MPa for 5 min, there were apparent damages visualized by SEM. The authors compared the alteration in the cell appearance to the cellular twist of cells. After HHP treatment, crushed and disintegrated cells were observed. Moreover, a progressive increase in the number of pimplelike lesions and swellings occurred. The results obtained through TEM also confirmed intracellular damages of *L. monocytogenes* cells. However, these changes appeared to be significantly different than those spotted in our experiment. The opposite phenomenon was achieved by Basaran-Akgul et al. [50] for a cocktail of three *L. innocua* strains visualized by SEM. Pressurization under 414 MPa and 517 MPa at 20 °C did not affect *L. innocua* cells. The same observations were confirmed by Ritz et al. [51]. They reported that the application of 400 MPa for 10 min did not disturb significantly membranes of *L. monocytogenes* CIP 103575 suspended in citrate buffer, visualized by SEM. Individual spotted changes were the bud scars on the cell surface. In the past couple years, different species and strains of *Listeria* sp., as a Gram-positive bacteria representative, were analyzed regarding the response to pressure treatment. The type of morphological changes differed among the *Listeria* genus in the mentioned publications. Although Gram-positive bacteria are known to be more resistant to pressure, the extent of triggered cell injuries may vary significantly. This phenomenon should be taken into account with the choice of pressure processing conditions, especially for products that carry a high risk of *Listeria* sp.

### 3.3. Physiological State of Cells Assessed by EFM

The physiological state observation of tested strains, evaluated by EFM, is shown in Figure 1, Figure 2, Figure 3 and Figure 4. Images obtained for all control samples demonstrated homogeneous fluorescing green populations of cells (SYTO9-stained). That is the evidence for the cells’ viability and their unaffected cell envelopes. As expected, cells occurred single-handedly, not in conglomerates. Staining HHP-treated samples showed a heterogeneous population. Both fluorescent green and fluorescent red cells were detected in all populations of tested strains. Opposite to control samples, cell aggregates were observed. The presence of red cell staining confirmed cell wall deterioration. Presumably, red cell aggregates represented a mixture of dead and injured cells. Due to the fact that cell conglomerates occurred, distinguishing single cells was impossible. Therefore, the calculation of the surface area of live and dead cells separately was carried out. The percentage ratio of live and dead (L/D) cells is shown in Figure 5 In the case of the *L. innocua*-tested strains, the L/D ratio was significantly higher for a collection strain (*p* < 0.05). There were 69% of viable cells and 31% of injured cells. In turn, the *L. innocua* wild-type strain characterized 55% and 45% of viable and injured cells, respectively. No significant differences between L/D ratios were reported for *E. coli* strains. There were around 40% and 60% of viable and injured cells in the populations, respectively. Prieto-Calvo et al. [11] analyzed the membrane integrity of two *E. coli* strains after the application of pressure (300 MPa and 600 MPa for 5 min) by EFM. They reported that the vast majority of HHP-treated cells were stained when exposed to PI, for both strains and both pressure levels. These results correspond to their TEM observations, in which the disruption and detachment of cellular envelopes were spotted. In turn, Huang et al. [49] examined cell membrane damages of *L. monocytogenes* BCRC 15354 after pressure application in a range from 250 MPa to 450 MPa. The increased uptake of PI corresponded to the increased value of pressure. In a sample treated with 450 MPa, the PI uptake increased 4.8 times in comparison with a control sample. Kimura et al. [52] detected injuries of *E. coli* ATCC25922 after HHP (400 MPa, 500 MPa, and 600 MPa after 5 min in 25 °C) by flow cytometry with the usage of set fluorescent dyes: propidium iodide and SYTO9. Just after pressure application, the profile of the population indicated only death and injured cells. However, the recovery of sublethally injured cells was perceived, after 24 and 48 h of incubation. The increase in cells in the living cell region emerged. Moreover, the PI/SYTO^®^ 9 emission signal of the injured cells was stronger than that of dead cells. Additionally, they reported that the population of the living cells detected in FCM analysis mostly corresponded to the results obtained for the plate assay. The abovementioned observation is similar to our study. Ritz et al. [51] showed results obtained for *L. monocytogenes* CIP 103575 after HHP treatment (400 MPa). The membrane integrity of tested bacteria was investigated with a PI fluorochrome by flow cytometry. A heterogeneous population after staining was observed. As a consequence of high-pressure treatment, a significant portion of cells was injured, which suggests that their membranes were seriously damaged. In this experiment, we proved that HHP’s effect on microbes is the appearance of internal structural damages. Although membrane permeabilization has been postulated as a major factor in the HHP-induced inactivation of microbes, the cell membrane of tested strains seems to remain intact beyond the *L. innocua* wild-type strain. Nevertheless, pressure triggered some physical damages. This was confirmed by EFM with staining analysis. PI can only enter cells via the membranes of injured or dead cells, which results in red fluorescence emission. Different types of staining analysis were performed in the abovementioned studies for the evaluation of the physiological state of HHP-injured bacteria. During this study, we tested the Backlight LIVE/DEAD vitality kit, as a quick method for injured cells. EFM analysis corresponds to observations obtained by TEM and SEM. Both the percentage ratio as well as the type and extent of morphological changes in the case of both *E. coli* strains were similar. In turn, the *L. innocua* wild-type strain, induced by pressure, was characterized with greater changes in internal cell organization than the *L. innocua* collection strain. It correlates with the results obtained by EFM, where the percentage of *Listeria* strain viable cells differed significantly.

## 4. Conclusions

The development of HHP technology has increased in recent years, and the extended diversity among food products has been reported. Nevertheless, this nonthermal treatment still has limitations. It has been demonstrated that changes in the cell morphology after pressure treatments were dependent upon the species [47], and the degree of damage may differ cell-by-cell [52]. To attain the quality and safety standards of high-pressure-processed foods, particular attention should be paid to the potential presence of sublethally injured cells. Research efforts have been made to clarify the relationship between sublethal injuries and products spoilage during long-term storage [53]. However, further studies are crucial to understanding the characteristics of HHP-injured bacteria and physiological changes of individual cells. The zone between lethal and sublethal injuries is probably a thin line. Moreover, the physiological state after treatment is commonly assessed only at the population level, while the presence of any subpopulation remains undetected. The microscopic approach enabled the analysis of both single cells and the whole population in terms of sublethal injuries. That mentioned above may support international standards in quality control laboratories.

## Figures and Tables

**Figure 1 foods-10-02940-f001:**
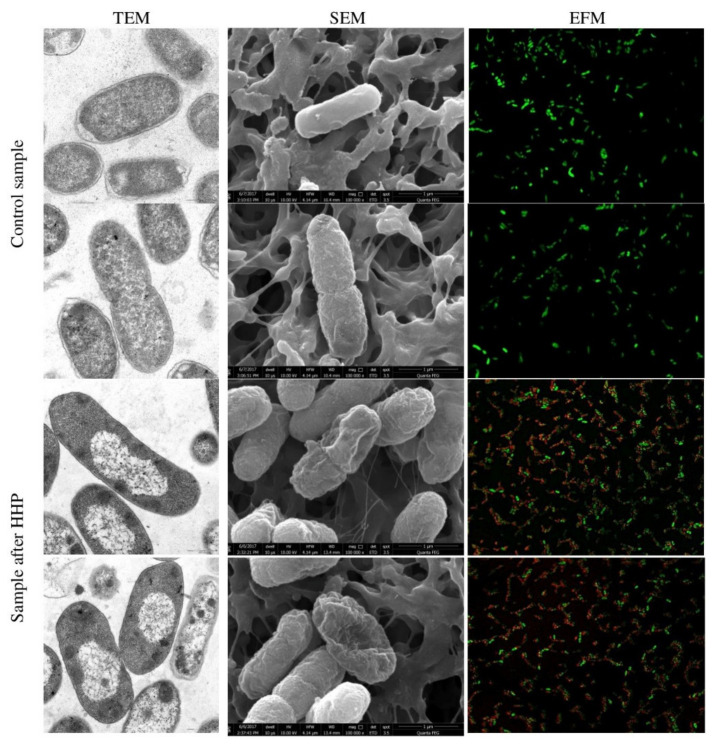
Representative images of *E. coli* ATCC 7839 before and after HHP treatment (500 MPa for 5 min). TEM micrographs scale bar, 0.5 µm. SEM micrograph shows bacteria at 10,000 times magnification. Images were made at an accelerating voltage of 10.0 kV with the use of an ETD. Uptake of Syto^®^ 9 (emission of green fluorescence) and PI (emission of red fluorescence) included in Live/Dead BacLightTM viability kit visualized by EFM.

**Figure 2 foods-10-02940-f002:**
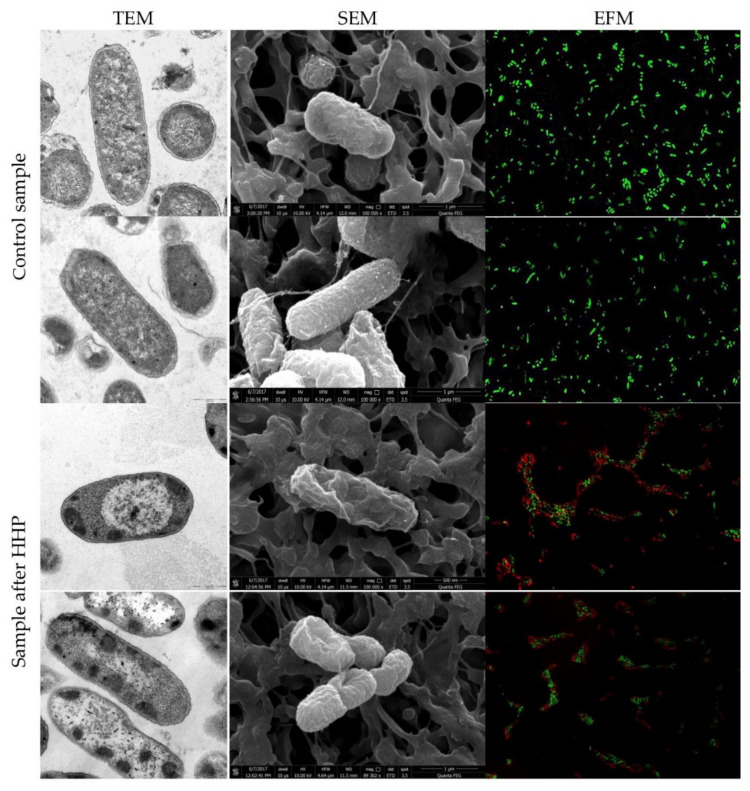
Representative images of *E. coli* 61/14 before and after HHP treatment (500 MPa for 5 min). TEM micrographs scale bar, 0.5 µm. SEM micrograph shows bacteria at 10,000 times magnification. Images were made at an accelerating voltage of 10.0 kV with the use of an ETD. Uptake of Syto^®^ 9 (emission of green fluorescence) and PI (emission of red fluorescence) included in Live/Dead BacLightTM viability kit visualized by EFM.

**Figure 3 foods-10-02940-f003:**
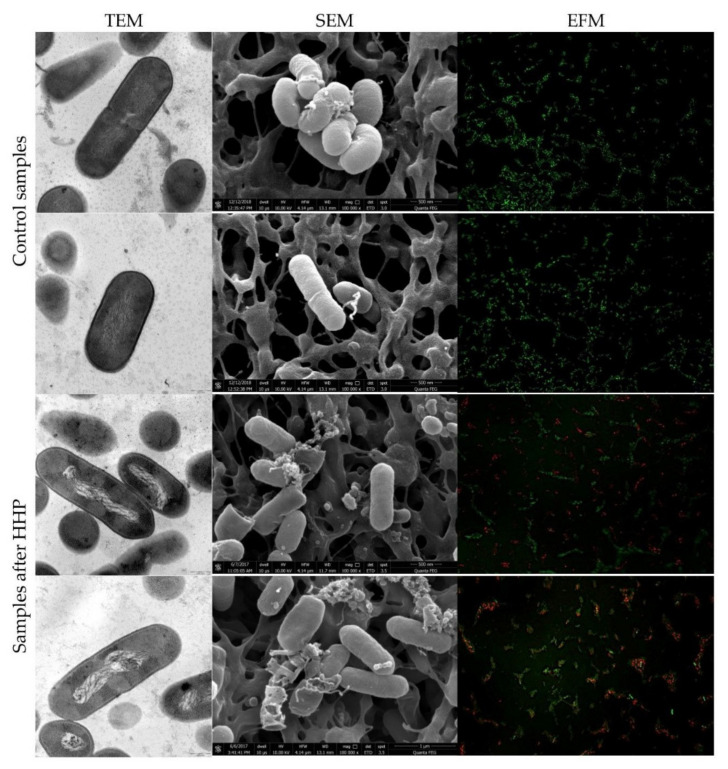
Representative images of *L. innocua* CIP80.11T before and after HHP treatment (400 MPa for 5 min). TEM micrographs scale bar, 0.5 µm. SEM micrograph shows bacteria at 10,000 times magnification. Images were made at an accelerating voltage of 10.0 kV with the use of an ETD. Uptake of Syto^®^ 9 (emission of green fluorescence) and PI (emission of red fluorescence) included in Live/Dead BacLightTM viability kit visualized by EFM.

**Figure 4 foods-10-02940-f004:**
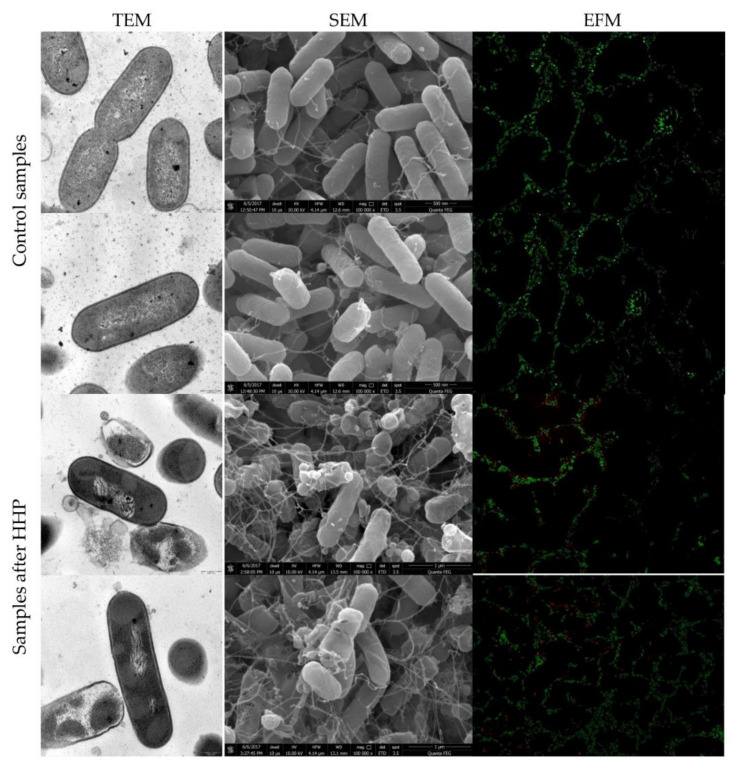
Representative images of *L. innocua* 23/13 before and after HHP treatment (400 MPa for 5 min). TEM micrographs scale bar, 0.5 µm. SEM micrograph shows bacteria at 10,000 times magnification. Images were made at an accelerating voltage of 10.0 kV with the use of an ETD. Uptake of Syto^®^ 9 (emission of green fluorescence) and PI (emission of red fluorescence) included in Live/Dead BacLightTM viability kit visualized by EFM.

**Figure 5 foods-10-02940-f005:**
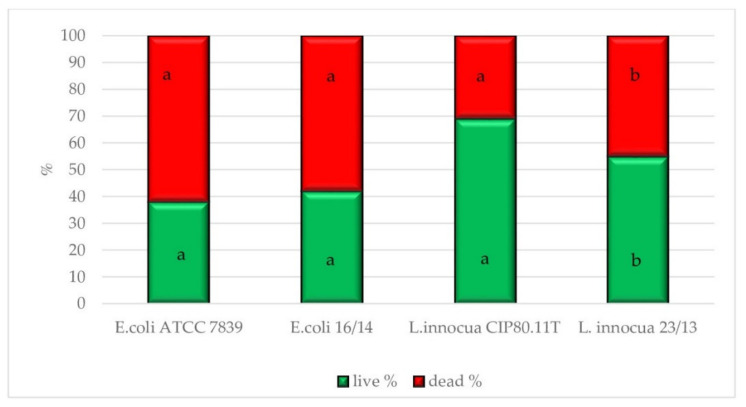
The percentage ratio of live and dead cells (with membrane permeabilization) after HHP process (*E. coli* strains—500 MPa/5 min and *L. innocua* strains—400 MPa/5 min). Results were calculated for 10 photographs of each strain. All experiments were performed twice. a, b values: live or dead cells denoted with the same letter are significantly different (*p* < 0.05).

## Data Availability

The data presented in this study are available on request from the corresponding author. The data are not publicly available, due to privacy.

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
