# Peer review of "High-Pressure-Induced Sublethal Injuries of Food Pathogens—Microscopic Assessment"

_foods, 2021, doi:10.3390/foods10122940_

Round 1

Reviewer 1 Report

The manuscript investigates changes in morphology of E. coli and L. innucua after treatment with high hydrostatic pressure used as a method for food preservation. The presented approach is based on the usage of microscopy techniques. The manuscript is well organized and the materials and method are described in details providing all necessary information.

The main drawback of the manuscript is that the it is not fully clear what is the novelty in the manuscript. In the results and discussion, authors mainly compare their results with already published ones but the discussion of the significance of the obtained results is scarce.

Also, the discussion could benefit a short mention and comparison of different effects of HHP on gram (-) and gram (+) bacteria since, I presume, the tested bacterial strains were chosen (among others) as representatives of each bacteria type.

The tittle should be shortened. It is not clear what the “..as a potential risk for foods…” is reffering to. It is, in my opinion unnecessary. Instead, authors could only write “…of food pathogens -….”

In figure legends of figures 1-4, the description for EFM part is missing and should be added.

In the introduction part, some sentences should be rewritten since they lack in clarity, mainly due to the difficulties with English language. For example:

Lines 50-51 the word order should be “….the EM has played the key role.”

Line 53 what is “cell components” referring to, written like this in the sentence?

Line 59-60, what is meant by the sentence “Obtained by…..”

Lines 78-79 it should state “….determinants…such as…”

In Materials and methods, the some numbers should be written in subscript lines 197, 198, H2O OsO4) or superscript (line 182, 107 CFU)

Author Response

I enclosed answers to the Reviewer’s 1 comments:

Point 1: The main drawback of the manuscript is that it is not fully clear what is the novelty in the manuscript. In the results and discussion, authors mainly compare their results with already published ones but the discussion of the significance of the obtained results is scarce.

Response 1: Two last paragraphs on the “Introduction” part have been modified to a better explanation of the necessity of this research. The arguments of the novelty have been added to the text.

Point 2: Also, the discussion could benefit from a short mention and comparison of different effects of HHP on the gram (-) and gram (+) bacteria since, I presume, the tested bacterial strains were chosen (among others) as representatives of each bacteria type.

Response 2: A few sentences regarding the different effects of HHP on the gram (-) and gram (+) bacteria have been added to the “Results and Discussion”section.

Point 3: The title should be shortened. It is not clear what the “..as a potential risk for foods…” is referring to. It is, in my opinion unnecessary. Instead, authors could only write “…of food pathogens -….”

Response 3: The title has been changed accordingly to the reviewer's recommendation.

“The potential risk for foods” was written within the meaning of possibilities to recover of sublethally injured cells in the food products. This manuscript focused on the range of sublethal injuries in the pathogens population. In our previous work, we proved, that the propagation and recovery of the pathogen cells, which were previously injured by pressure occurred during 28 days of storage. Therefore, different types of microscopic methods were used to determine the changes in the cells after high pressure and for a better understanding of bacteria cells' response to HHP.

Point 4: In figure legends of figures 1-4, the description for EFM part is missing and should be added.

Response 4: The description of EFM has been added in figure legends of figures 1-4: “ Uptake of Syto®9 (emission of green fluorescence) and PI (emission of red fluorescence) included in Live/Dead BacLightTM viability kit visualized by EFM.”

In the introduction part, some sentences should be rewritten since they lack clarity, mainly due to the difficulties with the English language. For example:

Point 5: Lines 50-51 the word order should be “….the EM has played the key role.”

Response 5: The abovementioned has been changed accordingly to the reviewer's recommendation.

Point 6: Line 53 what is “cell components” referring to, written like this in the sentence?

Response 6: “Cell components” means membranes, genome, ribosome. It has been added to the manuscript.

Point 7: Line 59-60, what is meant by the sentence “Obtained by…..”

Response 7: This sentence has been removed from the text. Unfortunately, it was an incomplete sentence from the previous version of the manuscript.

Point 8: Lines 78-79 should state “….determinants…such as…”

Response 8: The abovementioned has been changed accordingly to the reviewer's recommendation.

Point 9: In Materials and methods, some numbers should be written in subscript lines 197, 198, H2O OsO4) or superscript (line 182, 107 CFU)

Response 9: Numbers have been changed and rewritten in subscript and superscript accordingly to the reviewer's recommendation.

Reviewer 2 Report

The manuscript provides valuable information about the impact of HHP on the structural aspects of pathogenic bacteria using different microscopic methods. However, the paper needs correction through the manuscript to improve the writing. Also, the sentences can be shortened and make the manuscript concise with the same information. There are a few things through the manuscript that need to be corrected:

There are a few misspellings noticed in the manuscript. For example, the correct spellings are “assessment” in the title, “while” in line 28

Line 12: please delete “it was reported”

Line 19: please explain what is minor HHP?

Line 20: please replace “with the usage of” with “using”

Line 105-106: it can be deleted since it is the repetition of previous lines.

Why was the HPP applied just for 5 minutes? Are there any reasons behind it?

Line 349-350: how do you distinguish dead and injured cells precisely?

In figure 4, the cells affected by HHP shown by SEM have more small branches between cells compared to non-treated cells? What is that for?

Author Response

I enclosed answers to the Reviewer’s 2 comments:

Point 1: The manuscript provides valuable information about the impact of HHP on the structural aspects of pathogenic bacteria using different microscopic methods. However, the paper needs correction through the manuscript to improve the writing. Also, the sentences can be shortened and make the manuscript concise with the same information.

Response 1: Major changes has been made in the manuscript.

There are a few things through the manuscript that need to be corrected:

Point 2: There are a few misspellings noticed in the manuscript. For example, the correct spellings are “assessment” in the title, “while” in line 28

Response 2: The abovementioned has been changed accordingly to the reviewer's recommendation.

Point 3: Line 12: please delete “it was reported

Response 3: The abovementioned has been deleted accordingly to the reviewer's recommendation.

Point 4: Line 19: please explain what is minor HHP?

Response 4: Minor HHP means pressure above 200 MPa and below 600 MPa. On the industrial scale, the maximum HHP pressure parameter is 600 MPa, which causes the inactivation of the majority of vegetative microorganisms. In turn, it has been reported that 200 MPa pressurization is not successful in any injuries. The usage of such pressure parameters aimed at induction of the highest level of sublethal cell injuries in the bacterial populations. These process conditions were chosen based on our earlier studies.

The “minor” has been removed from the text. This term is not often used in publications.

Point 5: Line 20: please replace “with the usage of” with “using”

Response 5: The abovementioned has been changed accordingly to the reviewer's recommendation.

Point 6: Line 105-106: it can be deleted since it is the repetition of previous lines.

Response 6: The abovementioned has been changed accordingly to the reviewer's recommendation. Moreover, the two last paragraphs on the “Introduction” part have been modified to a better explanation of the necessity of this research.

Point 7: Why was the HPP applied just for 5 minutes? Are there any reasons behind it?

Response 7: As a preliminary experiment, we studied the effect of HHP on tested species in a range of pressure 200-500MPa up to 10 minutes at ambient temperature. A short processing time (<5 or 10 min) is desired since it has a significant effect on the economics of HHP processing. The next step that was done was the screening analysis to choose the parameters, which induce the highest level of sublethal injury of those bacterial strains. The results showed that pressure of 400 MPa for 5 minutes triggers sublethal injuries of Listeria innocua strains while extending the parameters inactivates these bacteria. In turn, Escherichia coli strains were sublethally injured under the pressure of 500 MPa for 5 minutes. This manuscript is a continuation of the abovementioned work. We focused here on the range of sublethal injuries in the pathogens population. Therefore, different types of microscopic methods were used to determine the changes in the cells after high pressure and for a better understanding of bacteria cells' response to HHP.

Point 8: Line 349-350: how do you precisely?

Response 8: The method of distinguishing dead and injured cells way of has been described in the “Introduction” part of the paragraph regarding EFM.

Point 9: In figure 4, the cells affected by HHP shown by SEM have more small branches between cells compared to non-treated cells? What is that for?

Response 9: These small branches presumably came from injured L. innocua cells, where the leakage of cell components occurred. There are visible in both the control and treated sample. Probably this might be the result of the sample preparation step. It was mentioned in the Results and Discussion section.

Round 2

Reviewer 1 Report

The authors have succesfully adressed all the comments from my prevoius review therfore I reccomend this manucript to be accepted for publication.